



# Inundation prediction in tropical wetlands from *JULES-CaMa-Flood* global land surface simulations

Toby R. Marthews[1], Simon J. Dadson[1,2], Douglas B. Clark[1], Eleanor M. Blyth[1], Garry Hayman[1], Dai Yamazaki[3], Olivia R. E. Becher[2], Alberto Martínez-de la Torre[1,4], Catherine Prigent[5], Carlos Jiménez[6]

[1]UK Centre for Ecology and Hydrology (UKCEH), Maclean Building, Wallingford OX10 8BB, U.K.
[2]School of Geography and the Environment, University of Oxford, South Parks Road, Oxford OX1 3QY, U.K.
[3]Institute of Industrial Science, University of Tokyo, 4 Chome-6-1 Komaba, Meguro City, Tokyo 153-8505, Japan
[4]Meteorological Surveillance and Forecasting Group, DT Catalonia, Agencia Estatal de Meteorología (AEMET), Barcelona, Spain
[5]CNRS, Laboratoire d'Etude du Rayonnement et de la Matière en Astrophysique et Atmosphères (LERMA), Observatoire de Paris, 61 avenue de l'Observatoire, 75014 Paris, France
[6]Estellus, 93 Boulevard de Sébastopol, 75002 Paris, France

*Correspondence to*: Toby R. Marthews (tobmar@ceh.ac.uk)

**Abstract.** Wetlands play a key role in hydrological and biogeochemical cycles and provide multiple ecosystem services to society. However, reliable data on the extent of global inundated areas and the magnitude of their contribution to local hydrological dynamics remain surprisingly uncertain. Global hydrological models and Land Surface Models (LSMs) include only the most major inundation sources and mechanisms, therefore quantifying the uncertainties in available data sources remains a challenge. We address these problems by taking a leading global data product on inundation extents (*GIEMS*) and

matching against predictions from a sophisticated global hydrodynamic model (*CaMa-Flood*) that uses runoff data generated from the *JULES* land surface model. The ability of the model to reproduce patterns and dynamics showed by the observational product is assessed in a number of case studies across the tropics (including the Sudd, Pantanal, Congo and Amazon), which show that it performs well in large wetland regions, with a good match between corresponding seasonal cycles. However, at finer spatial scale, water inputs (e.g. groundwater inflow to wetland) may become underestimated in comparison to water

outputs (e.g. infiltration and evaporation from wetland); or the opposite may occur, depending on the wetland concerned. Additionally, some wetlands display a clear spatial displacement between observed and simulated inundation as a result of over- or under-estimation of overbank flooding upstream. This study provides timely data that can contribute to our current ability to make critical predictions of inundation events at both regional and global levels.

## 1 Introduction

Wetlands and other inundated areas make up 6-8% of the terrestrial ice-free land surface (Junk et al., 2013; Mitsch and Gosselink, 2000, 2015). However, this percentage greatly underestimates their importance to the global climate system (WMO, 2019) and to human society (Mitsch and Gosselink, 2000). Wetlands, including peatlands (bogs and fens), mineral soil



wetlands (swamps and marshes) and seasonal or permanent floodplains (Saunois et al., 2020), play a key role in hydrological and biogeochemical cycles, are home to a large part of global biodiversity and provide value to human society in the form of multiple ecosystem services (Junk et al., 2013). Most significantly, wetlands and other inundated areas:

(i) Provide a spectrum of ecosystem services to human society including filtering of pollutants, maintenance of buffers against flood damage, reduction of soil erosion, biodiversity protection and recreational opportunities (Junk et al., 2013; Maltby and Barker, 2009; Mitsch and Gosselink, 2015);

(ii) Are the most significant natural source of atmospheric methane ($CH_4$), contributing 20-31% of global emissions of this highly potent greenhouse gas (Saunois et al., 2020)

and (iii) Mediate latent heat exchange between the atmosphere and the land surface, thereby greatly affecting the occurrence of deep convection and meso-scale precipitation systems (Prigent et al., 2011; Taylor, 2010; Taylor et al., 2018), with implications for the availability of freshwater resources (WMO, 2019).

## 1.1 Inundation extent

Inundation extent is a key impact variable related to wetland dynamics produced by hydrological models, which is calculated from a sequence of water balance calculations carried out over the course of the water cycle (at canopy level, ground level, etc.) (Hewlett, 1982; Sutcliffe, 2004). Precipitation received at the land surface is divided at the top of any vegetation canopy (canopy interception, dividing into canopy storage, throughfall and canopy evapotranspiration, e.g. Best et al. (2011)) and then again at the ground surface (dividing into infiltration (to soil water and drainage into groundwater), soil evaporation, surface ponding and lateral displacement). Heavy or persistent precipitation events may cause *surface water (pluvial) flooding* (= high levels of surface ponding or increased lateral displacement), resulting in higher runoff into local water courses. Once contained in water channels, most water flows along the river network to the ocean (*river routing*), but high river flows may exceed channel capacity downstream, producing an areal extent of inundated water (*overbank inundation*). Land surface inundation, if it occurs, is greater or lesser as a result of a balance between all of these factors.

Globally, we consider wetlands defined in the widest sense of any permanently or temporarily inundated area outside permanent water bodies (Ramsar, 2016). Wetlands may be divided according to their hydrotopographical context (Wheeler and Shaw, 1995) into *groundwater-maintained* or *groundwater-fed wetlands*, where the effects of groundwater dominate over other processes (e.g. fens, *depressional wetlands* of USEPA (2002) or the *non-flooded wetlands* of Miguez-Macho and Fan (2012)), and *fluvial inundation-maintained wetlands*, where their existence depends primarily on their proximity to a water course that regularly overtops its banks (e.g. igapó and várzea forests of the Brazilian Amazon, Pires and Prance (1985)). Seasonally-varying levels of inundation are primarily dependent on upstream precipitation and how this translates into these two forms of inflow, and secondarily on the ambient rates of evaporation and infiltration (Clark et al., 2015; d'Orgeval et al., 2008; Marthews et al., 2019). Further classification of wetlands in terms of vegetation or substrate is not required for our study (but see Wheeler and Shaw (1995), USEPA (2002), Gerbeaux et al. (2018) and Ramsar (2016)). The characterisation of the





variation of inundation as a result of the cycles and variability of all these processes is the primary challenge in simulating and

predicting inundation (Yamazaki et al., 2011).

## 1.2 Uncertainty in observations

Much of the uncertainty in the magnitude of important fluxes related to wetlands, is attributable to the wide range of estimates of global inundated areas (Aires et al., 2018; Hu et al., 2017; Melton et al., 2013; Parker et al., in prep. 2020; Pham-Duc et

al., 2017; Tootchi et al., 2019). The importance of reducing this uncertainty has long been known from the perspective of policymakers concerned with implementing natural flood management plans (Dadson et al., 2017; Junk et al., 2013; Moomaw et al., 2018) or working in regions where water resources are under threat (Mitsch and Gosselink, 2000; Vörösmarty et al., 2010), but over the last decade this has additionally been recognised more widely in the scientific community in terms of predictions of climate change (Thirel et al., 2015; Zhao et al., 2017). Unfortunately, progress has been relatively slow because

of the challenge of simultaneously improving both our observations and our predictions of global inundation extents.

Assessing the precise extent of natural wetlands and other inundated areas from remote sensing remains challenging across large regions (Dutra et al., 2015), especially in the context of constraining process models that produce estimates of wetland extent (see discussion in Saunois et al. (2020)). Observational uncertainty depends on the form of inundation (e.g. deep *vs.* shallow, colder *vs.* warmer water) and ambient conditions (e.g. flooding occurring during a storm under cloud cover

*vs.* from snowmelt under clear conditions, or occurring during night *vs.* day hours). Additionally, there are the more general uncertainties in remote sensing products stemming from thresholding assumptions and/or compositing (e.g. see Liang and Liu (2020)). Uncertainty in inundation extent observations continues to be an issue in any study based on remote sensing data, e.g. this uncertainty has recently been shown to be the most significant factor in global $CH_4$ budget uncertainty (Parker et al., in prep. 2020).


## 1.3 Uncertainty in model predictions

Many hydrologic models exist that are capable of simulating flood inundation, however these models differ greatly in their sophistication, the breadth of water cycle processes included and their optimal scale of application (Beck et al., 2017; Clark et al., 2017; Clark et al., 2015; Davison et al., 2016; Dutta et al., 2000). Inundation models seldom include all forms of inundation

and hydrological processes (Clark et al., 2015; Davison et al., 2016), and the absence of even one process can lead to significant underestimation of inundation extent (e.g. as found by Parker et al. (2018) for the process of overbank inundation). The storage and conveyance of water in lakes, floodplains, groundwater and river channels, especially, is generally simulated only with relatively high uncertainty in the current generation of land surface models (LSMs) (Marthews et al., 2020; Marthews et al., 2019).

Most hydrological models are run uncoupled from the atmosphere and are therefore reliant on the availability of good precipitation and other atmospheric driving data. Uncertainties in the precipitation driving data may often be very significant and larger than the total uncertainty inherent within the model being run (Marthews et al., 2020). Previous studies have



attempted to validate global hydrology models against global hydrology products (e.g. Beck et al. (2017) based on the WRR1 configuration (see Schellekens et al. (2017), now updated to WRR2 by Fink and Martínez-de la Torre (2017)) and also see (Decharme et al., 2012; Gedney et al., 2014; Stacke and Hagemann, 2012; Sterk et al., 2020; Yamazaki et al., 2011)). However, many such studies evaluated only runoff or river flow against corresponding models (e.g. Zhao et al. (2017)), without consideration of the areal extent of inundation as we have done in this study.

### 1.4 Model and study area selection

The global flood simulation model *CaMa-Flood* was selected for our predictions of inundation extents because of its sophistication and the fact that it is already widely-used (Hoch and Trigg, 2019; Zhao et al., 2017). *CaMa-Flood* is the only open-source global river routing model that is based on the local inertial approximation of the Saint Venant equations (Bates et al., 2010; Dutta et al., 2000; Fassoni-Andrade et al., 2018; Yamazaki et al., 2013), which takes into account the backwater effects of downstream elements, i.e. the possible reversal of flow in particular reaches upstream from e.g. lakes, tributaries, estuaries (Hidayat et al., 2011). By including these effects, *CaMa-Flood* is able to produce a much better characterisation of many wetlands whose dynamics are dominated by surface water inundation.

*CaMa-Flood* requires runoff data for its simulations, which we obtained from runs of the UK land surface model *JULES* carried out previously through the EU *eartH2Observe* project (Schellekens et al., 2017; Sterk et al., 2020). We chose to use this *JULES*-based dataset because uncertainty in water cycle quantities for *JULES* were comparable to any other equivalent land surface model (Marthews et al., 2020) and because streamflow and runoff data produced by this model have already been validated at a global level (Martínez-de la Torre et al., 2019). Additionally, through using these models, our results can contribute to the current effort to include global flood inundation in the *JULES* model itself (Dadson et al., submitted 2021; Lewis et al., 2019; Lewis et al., 2018).

Our comparison of model and observational data was carried out over the whole tropical zone (23.5°S to 23.5°N, excluding small oceanic islands) at a resolution of 0.25° in both latitude and longitude (Fig. 1). We have taken a case study approach (Table 1), where our wetland areas were selected on the basis of being the largest extant global wetlands, with two limitations. Firstly, we avoided regions with significant inundation on frozen and partially-frozen land because *GIEMS* does not account for frozen water and areas with significant snowfall are systematically masked as well (Prigent et al., 2007). Secondly, coastal or tidal wetlands were also avoided because their interactions with the ocean cannot currently be simulated by *JULES* or *CaMa-Flood*. Because of the preponderance of coastal occurrence across subtropical and temperate wetlands (Gumbricht et al., 2017; Melton et al., 2013), with these two limitations all remaining large wetlands were in the tropical zone (23.5°S to 23.5°N).

In this study, we ask the following questions:

(1) How well can the *CaMa-Flood* model, driven by *JULES* runoff data at 0.25° resolution, simulate observed global inundated extents, as given by *GIEMS* satellite-based data?



(2) Can an improved match between observed and predicted inundation be obtained by simple transformations, e.g. removing low/high observed values or adding a constant to all predicted inundation fractions?

(3) Are these simple transformations dependent on spatial scale (e.g. regional *vs.* subcontinental)?

Answering these questions will highlight both the strengths and weaknesses of the *JULES-CaMa-Flood* approach to global inundation prediction and indicate possible directions where improvements may be made in modelling predictive capability in global wetlands.

## 2 Methods


Observed and simulated inundation extents were compared at a global resolution of 0.25° x 0.25° (approximately 25 km x 25 km at the Equator).

### 2.1 Observed inundation extents

Observational data on monthly global inundation fraction were obtained from the Global Inundation Extent from Multi-Satellites database version 2.0 *GIEMS-2* (Prigent et al., 2020), which is considered to be one of the best, widely-available global products of inundation extents and captures water under vegetation very well (Hu et al., 2017; Pham-Duc et al., 2017). Data were regridded to a regular spatial resolution of 0.25° x 0.25° to enable comparison with model outputs.

*GIEMS* is mainly derived from passive microwave observations (Special Sensor Microwave/Imager (SSM/I) and
SSMIS), with the help of active microwave and visible and near infrared reflectance observations (Advanced Very High Resolution Radiometer (AVHRR)) to eliminate ambiguities in surface water detection and to account for the potential contribution of vegetation (Prigent et al., 2020; Prigent et al., 2007). *GIEMS* can detect inundation of both natural wetland and irrigated agricultural areas. Frozen surfaces are excluded. In unfrozen areas, the accuracy of *GIEMS* has been comprehensively verified (Papa et al., 2010; Papa et al., 2006) and it is a very widely used remote sensing product (e.g. (Taylor et al., 2018;
Zhang et al., 2016)), therefore it forms an appropriate benchmark dataset for global modelling studies.

### 2.2 Simulated inundation extents

Model-derived inundation extents were produced by a sequentially executed run of two models referred to here as *JULES-CaMa-Flood*. Firstly, predictions of land surface runoff were obtained from the UK land surface model *JULES*
https://jules.jchmr.org/ (Best et al., 2011; Clark et al., 2011) by accessing simulations carried out previously through the EU *eartH2Observe* project (Marthews et al., 2020; Schellekens et al., 2017; Sterk et al., 2020). A validation of these runoff data and a description of the hydrological simulation approach and water balance calculations in *JULES* is given in Martínez-de la Torre et al. (2019).

Secondly, these runoffs were used to drive the flood inundation model *CaMa-Flood* v3.9.6a (version November 2019)
(Yamazaki et al., 2011; Yamazaki et al., 2009), to produce predictions of surface inundation at all points. *CaMa-Flood* was





run at a sub-daily timestep (timestep 1 min for runs; 1 day for driving data) and then the outputs were averaged to produce monthly data. *CaMa-Flood* was set to calculate river discharges and flow velocities using the local inertial equation along its river network map in order to include backwater effects (Bates et al., 2010; Yamazaki et al., 2013; Yamazaki et al., 2011). In order to compare more easily with observations on a regular grid, our *CaMa-Flood* simulations were in fully grid-based mode

rather than using irregularly-shaped catchments (Yamazaki et al., 2011; Yamazaki et al., 2009). *CaMa-Flood*'s options for bifurcating flows within the model were not activated for these simulations (Yamazaki et al., 2014) because we did not include coastal wetlands in our case studies (only in coastal wetlands would bifurcation occur at a spatial scale greater than our gridcell scale of approximately 25 km) and because we focus on water balance in our analysis (which should be negligibly affected by river braiding and other bifurcations).


### 2.3 Analysis

The period for which *eartH2Observe* and *GIEMS-2* data overlap is 1992-2014, so we used this period for all our analyses. All post-processing steps were carried out using NetCDF Operator (NCO) tools v.4.4.5 (Zender, 2008) and the statistical language environment R v.4.0.2 (R Core Team, 2020). For the R-based analyses, packages *maps*, *rgeos* (v.0.5-3), *GEOS runtime*

(v.3.8.0) and *rgdal* (v.1.5-12) were required. All code used in the analysis will be made available on request.

#### 2.3.1 Evaluation metrics

We applied the two most common efficiency statistics used in the context of river flow analysis: the Nash-Sutcliffe Efficiency (NSE) and Kling-Gupta Efficiency (KGE), both of which measure the alignment between model results and observations

(Table 2). KGE is based on a decomposition of NSE into its constitutive components (correlation, variability bias and mean bias) and addresses several perceived shortcomings in NSE (Knoben et al., 2019).

Our focus in this study is wetlands, therefore we excluded areas of very high inundation (permanent lakes and reservoirs, which were always 100% inundated in both observed and simulated data because of substitution from the GLWD (Lehner and Döll, 2004)) and also areas of continuously low or zero inundation (dry areas in the validation region, which

would also provide a constant match between observed and simulated areas, see e.g. (Bernhofen et al., 2018)). Our focus on variability measures ensured that our match statistics were dominated by the regular (seasonal) and irregular cycles occurring at points where inundation was not constant, i.e. wetland regions *sensu stricto*.

#### 2.3.2 Transforming inundation extents

When comparing the observed and simulated inundation extents, it appears to be the case that a certain amount of inundation is predicted by *JULES-CaMa-Flood* but is not observed by *GIEMS* (e.g. Sudd results, Fig. 1). Based on the data we have, it is not possible to be certain whether this 'low level' inundation shows some kind of bias towards overprediction on the part of the model, or perhaps the inundation is actually real but for some reason unobserved by *GIEMS* (see e.g. Liang and Liu (2020)



for a discussion on the limitations of the satellite-based sensors employed). In order to test this, during our analysis we posit a
nonzero, minimum level of inundation fraction *alpha_min* below which *GIEMS* always returns a zero result.

It is also possible that there is a maximum inundated fraction (here called *alpha_max*) above which *GIEMS* loses its
sensitivity (i.e. possibly *GIEMS* can differentiate well between 20% and 30% inundation, but not as reliably between 70% and
80%). This may possibly happen because vegetation canopy cover obscures inundation occurring beneath it, and the magnitude
of this effect will depend on canopy coverage and the density of the canopy concerned, among other factors (*GIEMS* is capable
of detecting some water under dense vegetation, but with high uncertainty, especially when the distribution of inundation
within the gridcell is highly skewed, i.e. small dry areas within a very wet gridcell or *vice versa*) (Prigent et al., 2020).

Finally, it may also be the case that our predictions of inundated fraction have a systematic bias (underestimation or
overestimation, on a gridcell-by-gridcell basis). In order to test this, we introduce a fraction *beta* which is added to all *CaMa-
Flood* outputs of *flooded fraction* (*fldfrc*). In summary, we can modify the *GIEMS* data and *CaMa-Flood* outputs according to
the simple transformations in Fig. 2 in order to investigate and quantify bias in both our simulated and observed data.

## 3 Results

Results are presented in a sequence of case study areas, beginning with the Sudd, Pantanal, Tonlé Sap, Inner Niger Delta and
Okavango wetlands before moving to the larger, subcontinental wetland complexes of the Central Amazon and the Congo
Cuvette. Straight comparisons between observations and model predictions of inundation show a complicated pattern of partial
overlap that is challenging to assess visually (Suppl. info). We therefore calculate spatial matching statistics across all case
study areas.

### 3.1 Inundation extent

*GIEMS* observations and *JULES-CaMa-Flood* predictions match very variably: monthly average inundation extent shows a
clear bias in most study wetlands, and in addition there is significant year-on-year variability (Fig. 3). However, the direction
of bias is not consistent between wetlands. Mapping pixel-based calculations of error (Normalised RMSE) and correlation
coefficient (Pearson's *r*) indicated that the correspondence between observed and simulated data is generally good (low RMSE)
and correlations are almost always positive (high *r*), however plots of RMSE and Pearson's *r* contained no information not
visible on the corresponding plots of NSE and KGE and are therefore not shown (because these metrics are modified versions
of those statistics, Table 2).

Nash-Sutcliffe and Kling-Gupta efficiency scores are most usually used in relation to discharge data, yielding
generally only one time series per catchment (see Suppl. Info), but in this study we have inundation estimates at every gridcell
and therefore it is possible to calculate efficiency on a pixel-by-pixel basis in each of our study areas (Fig. 4). Averaged
efficiency scores are generally high within the borders of the wetland itself, although lower in parts of the wetlands that have
the most dynamic flow regime.





However, these statistics are not capable of measuring some aspects of the flow regime that are important from the point of view of allowing us to divide out the different sources of inundation in our study wetlands. For example, the Inner

Niger Delta wetland shows apparent spatial displacement of inundation between observed and simulated: *GIEMS* reports negligible inundation north of 15.5°N in any month (a result broadly in line with the finer scale analysis of Bergé-Nguyen and Crétaux (2015)) even though *CaMa-Flood* predicts inundation reaching as far as Timbuktu at 16.5°N (Fig. 1). At this spatial resolution, 1° latitude should be easily resolved so this is a significant mismatch.

**3.2 Identifying an optimal transformation of GIEMS observations and JULES-CaMa-Flood predictions**

Varying the values of the three parameters *alpha_min*, *alpha_max* and *beta* (see Fig. 2), we searched for an optimal value of each that brought our observed and simulated data as close together as possible, in order to quantify and therefore help understand the discrepancy between our model result and the (uncertain) observations. By repeating the calculations that produced Fig. 4 for a range of reasonable parameter combinations of *alpha_min*, *alpha_max* and *beta*, the state space plots in

Fig. 5 were produced. The visible maxima on these state space plots provide a best estimate of the optimal values of these parameters, with these optima differing markedly between our wetland study areas (Table 1). A notably higher value for NSE or KGE for a particular combination of *alpha_min*, *beta* and *alpha_max* would identify a consistent bias in either the model predictions or the observations (or both).


**4 Discussion**

There has recently been significant progress in our understanding of wetlands and the roles they play in climate processes, land surface processes and their impacts on human society (IPCC, 2014; Mitsch and Gosselink, 2015; Moomaw et al., 2018; Saunois et al., 2020). However, even though the physics of flood inundation is relatively well-known (Bates et al., 2010;

Fassoni-Andrade et al., 2018; Yamazaki et al., 2013), many hydrological processes relevant to the representation of flooding in Earth system models remain poorly characterised at the high resolutions required to address issues of local and regional impact (Bierkens, 2015; Clark et al., 2015; Marthews et al., 2019; Zhou et al., in prep. 2020), including infiltration (Clark et al., 2015; d'Orgeval et al., 2008), and evaporation (d'Orgeval et al., 2008; Robinson et al., 2017) of flood waters, as well as groundwater effects (Clark et al., 2015).

In this study, we have simulated inundation extent at a spatial resolution high enough to resolve the major details of most major global wetlands. These results are potentially of great use to a wide audience of academic and non-academic users interested in the broad-scale impacts of environmental change on wetlands, especially where seasonal inundation affects water and energy fluxes in Earth system models. It is therefore appropriate to seek as robust a validation of these predictions as possible.




## 4.1 Comparing simulated and observed global inundated extents

We found that our simulated inundation extents (from the *CaMa-Flood* model, driven by *JULES* runoff data at 0.25° resolution) sometimes compared very closely to our observed data (from *GIEMS* satellite-based data), but at many points there were divergences (Fig. 1). For example, in the Sudd wetland, our model appears to over-predict inundation, whereas in the Pantanal
it appears to under-predict (Fig. 1). Can we explain this difference between *GIEMS* observations and our model predictions?

In order to investigate these divergences, we applied simple transformations to our data (Fig. 2) and the optimal values of the three parameters *alpha_min*, *alpha_max* and *beta* we found for each wetland provide robust explanations for observable differences. We found that our predictions of inundation extent could be improved at local or regional scale by simple transformations involving the three parameters *alpha_min*, *alpha_max* and *beta*. Moreover, in what follows we use our
diagnosis of these differences to highlight opportunities to improve the representation of physical processes in land-surface and large-scale hydrodynamic models.

We found evidence that *alpha_min* might generally take a nonzero value ~10% (Fig. 5b), indicating that *GIEMS-2* may be missing widely-distributed occurrences of low inundation within these wetlands, as suggested by previous studies (Prigent et al., 2007). Although we accept that *GIEMS* may underestimate low levels of inundation that occur outside wetlands
because of uncertainties in estimating inundation e.g. below intact forest canopies (although small in any particular location, these would sum to a significant missing term in regional and continental water budgets), however we believe that most of this percentage is simply indicating that *JULES-CaMa-Flood* overestimates inundation in wetland areas (which is then averaged out with the zero bias outside wetland areas).

We found no evidence to suggest that *alpha_max* should consistently take any value <1.0 for any of our wetlands
(Fig. 5; i.e. we found no evidence that the *GIEMS-2* inundation extents overestimated inundated fraction in gridcells where inundation covered a large percentage of the spatial cell)

We found high variation in the estimated value of *beta* for each wetland (Fig. 5b), i.e. adding a consistent constant fraction of inundation extent to all gridcells within the limits of each study wetland did indeed provide a closer match between observations and simulation, at least in the wetlands we considered in this study. Our interpretation of this is influenced by the
consideration that we know the *JULES-CaMa-Flood* model does not simulate several hydrodynamic processes that are known to have a great impact on inundation extent (e.g. evaporation of flooded areas). We suggest that the negative values of *beta_opt* in the Sudd and Inner Niger Delta show probable underestimation of hydrological output by *JULES-CaMa-Flood* (*water_out*). Conversely, the positive values of *beta_opt* in the Okavango show probable underestimation of hydrological input by *JULES-CaMa-Flood* (*water_in*).

The spatial displacement of inundation prediction downstream from observed inundation visible especially in our results for the Inner Niger Delta and the Sudd (Fig. 1) is a result of over- or under-estimation of overbank flooding upstream. If overbank flooding is underestimated in our simulation then the water within the river course (the Niger or White Nile, respectively, in these cases) will remain in the river and be taken downstream further than expected, producing a downstream wetland 'extension' that exists in the simulation results but not the observed (as we see in our *JULES-CaMa-Flood* outputs).






## 4.2 Implications for the hydrodynamic balance of wetlands

Wetlands exist as a balance between water input and water output, where we may define *water_in* = (channel + surface + subsurface inflow + local precipitation) and *water_out* = (infiltration + evaporation) (Fig. 5b) (i.e. a landscape-scale water balance, Sutcliffe (2004)). In order to understand these and other points of divergence between observation and prediction, we

need to understand this balance calculation in that particular wetland, and also assess what types of water bodies are represented in the simulated data (Zhou et al., 2020; Zhou et al., in prep. 2020).

The optimal parameter value derived in this study *beta_opt* may be understood as an index unique to each wetland that estimates the amount that is missing or underestimated in the overall wetland water balance. For example, *beta_opt* will be negative if evaporation and infiltration are being significantly underestimated by *JULES-CaMa-Flood* in this study area

(neither *JULES* nor *CaMa-Flood* explicitly models evaporation from inundated water in their present configurations). Conversely, *beta_opt* will be positive if e.g. groundwater inflow is being underestimated. Therefore, the value of *beta_opt* may be thought of as an estimate of how much *water_in* is underestimated by *CaMa-Flood* minus how much *water_out* is underestimated.

Categorising wetlands in terms of positive or negative *beta_opt* would be superficially similar to the division by Junk

et al. (2011) of South American wetlands into *fluvial* (wetlands that are predominantly maintained by river overbank inundation rather than by groundwater effects) and *interfluvial* wetlands (where groundwater effects dominate), however theirs was a distinction based on overall water balance rather than the balance of water input. In the context of our analysis here, we understand fluvial and interfluvial wetlands to mean ones where *water_in* is dominated by channel/surface flow or subsurface inflow, respectively. However, both fluvial and interfluvial wetlands may of course experience high evaporation rates (e.g. the

Inner Niger Delta) or high infiltration rates (based on underlying soil type) and therefore may occur either above or below the *y*=0 line in Fig. 5b.

## 4.3 Inundation at subcontinental and larger scales

Looking at subcontinental scales (the Amazon and the Congo) and larger scales (the three tropical zones), a number of

additional considerations become more important. As with all very large river basins, the inland reaches of the Amazon and the Congo are collectively enormous wetland complexes (Fig. 1), with some areas dominated by river flow and others by topographic factors (e.g. the "cuvette" of the Congo Cuvette indicates the whole subcontinent is approximately a shallow bowl). The same diagnosis of biases may be carried out over these larger areas, but our optimal value for *beta* generally converges closer and closer to the 'null' value *beta*=0.0 as larger and larger regions are considered (at least, for regions that

do not include significant coastal or permafrost areas). This is reasonable, because even the largest wetland areas are localised regions at this scale and therefore these optima will be averaged together with an increasing number of relatively *terra firme* gridcells (i.e. gridcells which experience little or no regular inundation) and, at the largest scales, with entire mountain ranges where little or no inundation occurs (either in our model or in the observations).



In addition, we should expect that *beta_opt* should converge to zero at the largest scales because we know that these models return reliable global estimates (Yamazaki et al., 2011), therefore from a global perspective the magnitude of values for a particular wetland or wetland complex should be understood as biases that are balanced out elsewhere. However, wetland-specific values nevertheless provide useful information about the inundation processes that dominate in those particular wetlands and allow us to improve our understanding of landscape-scale and continental-scale inundation hydrodynamics.

**4.4 Conclusions**

Simulations of inundation extent are important because they allow us to predict what will happen to globally-important wetlands in the future. Wetlands are known to be key nodes in the biosphere system in terms of vulnerability to climate change (Maltby and Barker, 2009; Mitsch and Gosselink, 2015). However, wetlands are highly dynamic landscape-level entities produced by the balance of a number of different water cycle processes acting together (Hewlett, 1982; Sutcliffe, 2004), not

all of which are yet represented in global hydrodynamic models (Yamazaki et al., 2013; Yamazaki et al., 2011).

Reducing uncertainty in predictions from large-scale inundation models has long been a prerequisite for their use in global Earth system models. In this study we have shown that a very reasonable and close match may be derived between *JULES-CaMa-Flood* model predictions of inundation extent and independent *GIEMS-2* global satellite-based observations of inundation. Differences do occur at regional scale in particular large wetlands, however, and these differences indicate clearly

the importance of incorporating into the modelling framework a better representation of the hydrological impacts of, especially, infiltration, evaporation and groundwater-fed inundation.

Improving our understanding of the dynamics of inundated areas and the role they play in the generation of land-atmosphere fluxes requires a better representation in general of wetlands within global land-surface and hydrodynamic models (Zhang et al., 2016). The results of this study point clearly towards the need for greater attention to be paid to hydrological

dynamics and water cycle processes within these models, which we expect to result in improved modelling predictive capability in global wetlands in the future. A firm focus on producing a better characterisation of hydrodynamics within this class of models will produce enormous positive returns in terms of our global capability to predict inundation and its global impacts and will make a welcome contribution to our preparedness for the impacts of future climate change (IPCC, 2014; Moomaw et al., 2018).


**5 Acknowledgements**

This work was supported by the Natural Environment Research Council through the National Capability *Hydro-JULES* project (grant number NE/S017380/1). TRM thanks colleagues at UKCEH for extremely useful discussions during the development

of this paper. This work used *JASMIN*, the U.K.'s collaborative data analysis environment (http://jasmin.ac.uk). GH acknowledges the support of NERC through the grant *The Global Methane Budget, MOYA* (NE/N015746/2)


**Table 1**: The wetland case study areas. Total tropical land area is approx. 56 000 000 km$^2$ (approx. 38% of total global land)

| Site | Location | Surface area |
|---|---|---|
| Neotropics | 23.5°S to 23.5°N, 110.4°W to 34.6°W | Approx. 18 000 000 km$^2$ land area (Malhi, 2010) |
| Amazon | The Central Amazon (Brazil, Colombia, Peru) 15.0°S to 7.0°N, 75.0°W to 47.0°W | Approx. 1 900 000 km$^2$ (Gedney et al., 2019; Yamazaki et al., 2011) |
| Pantanal | The Pantanal (Brazil, Bolivia, Paraguay) 22.0°S to 14.8°N, 61.1°W to 54.6°W | Varies up to 220 000 km$^2$ (Parker et al., 2018) |
| West Paleotropics | Tropical Africa and Arabia 23.5°S to 23.5°N, 17.6°W to 64.0°E | Approx. 21 000 000 km$^2$ land area |
| Niger Inland Delta | The Inner Niger Delta wetland (Mali) 13.6°N to 17.1°N, 5.2°W to 2.8°W | Varies up to 80 000 km$^2$ (Andersen et al., 2005; Balek, 1977; Bergé-Nguyen and Crétaux, 2015; Dadson et al., 2010; Haque et al., 2020) |
| Sudd | The Sudd (South Sudan) 4.5°N to 10.0°N, 28.0°E to 33.0°E | Varies up to 64 000 km$^2$ (Balek, 1977; Mohamed and Savenije, 2014; Sutcliffe and Parks, 1999; Tootchi et al., 2019), including the Bahr el Ghazal to the west and the Machar marshes to the east. |
| Congo | The Congo Cuvette Centrale (D. R. Congo, Congo-Brazzaville) 3.2°S to 3.6°N, 14.6°E to 25.2°E | Approx. 1 000 000 km$^2$ (Alsdorf et al., 2016; Balek, 1977; Betbeder et al., 2014) |
| Okavango | The Okavango Wetlands (Botswana) 24.0°S to 16.0°S, 19.0°E to 27.0°E | Varies up to 38 000 km$^2$ (the main delta NW of Maun varies up to 22 000 km$^2$ and the Makgadikgadi pans are an additional 16 000 km$^2$) (Milzow et al., 2009; Wolski et al., 2012).. |
| East Paleotropics | India to New Guinea 23.5°S to 23.5°N, 64.0°E to 153.5°E | Approx. 17 000 000 km$^2$ land area |
| Tonlé Sap | Tonlé Sap wetland (Cambodia) 11.6°N to 13.6°N, 103.0°E to 105.1°E | Varies up to 16 000 km$^2$ (Sithirith, 2015) |




**Table 2**: Efficiency metrics widely used in flood model assessment and forecast verification (Knoben et al., 2019). In all equations, $Q$ = flow variable (e.g. discharge) over time steps $t=1,..,T$. Subscripts "obs" and "sim" refer to observed and model-

predicted values, respectively, $\mu_{obs} = \overline{Q_{obs}}$ is the observation mean and $\sigma_{obs} = \sqrt{\frac{1}{N-1}\sum_t(Q_{obs}(t) - \overline{Q_{obs}})^2}$ is the standard

deviation (and similarly for $\mu_{sim}$ and $\sigma_{sim}$) and $r$ is the Pearson correlation coefficient between observed and simulated values.

| Evaluation metric | Equation | Description |
|---|---|---|
| Nash-Sutcliffe efficiency (NSE) [*] | $NSE = 1 - \dfrac{\sum_t\left(Q_{sim}(t) - Q_{obs}(t)\right)^2}{\sum_t(Q_{obs}(t) - \overline{Q_{obs}})^2}$ | Standard thresholds for NSE (but see Supp. Info):<br>1.0 = Perfect alignment<br>> 0.5 = Good alignment (Decharme et al., 2012; Knoben et al., 2019) (although some other authors specify >0.6, e.g. Martínez-de la Torre et al. (2019))<br>0.0 = No predictive skill (mean of observations provides as good an estimate as simulations)<br>< 0.0 = Increasing divergence between simulations and observations<br>Note that in this study points of very low inundation (dry areas *sensu* Bernhofen et al. (2018)) and very high inundation (permanent lakes and reservoirs) were removed before calculating NSE (because of the requirement to have at least some flow variability for the calculation), therefore our NSE values were slightly lower than usual. Our analysis rests on relative rather than absolute values of NSE, so our results are unaffected by this, but for clarity of comparison between sites we have used a consistent colour scale on all NSE plots based on the standard thresholds. |
| Kling-Gupta efficiency (KGE) [*, **] | $KGE = 1 - \sqrt{(r-1)^2 + \left(\dfrac{\sigma_{sim}}{\sigma_{obs}} - 1\right)^2 + \left(\dfrac{\mu_{sim}}{\mu_{obs}} - 1\right)^2}$ | Standard thresholds for KGE:<br>1.00 = Ideal model performance<br>> $(1-\frac{1}{\sqrt{2}}=)$ 0.29 = Good performance (Knoben et al., 2019)<br>$(1-\sqrt{2}=)$ -0.41 = No predictive skill (mean of observations provides as good an estimate as simulations; n.b. negative values above this threshold still indicate that a model is an improvement over the mean flow benchmark) (Knoben et al., 2019)<br>< -0.41 = Increasing divergence between simulations and observations<br>Note that in this study points of very low inundation (dry areas *sensu* Bernhofen et al. (2018)) and very high inundation (permanent lakes and reservoirs) were removed before calculating KGE (because of the requirement to have at least some flow variability for the calculation), therefore our KGE values were slightly lower than usual. Our analysis rests on |



relative rather than absolute values of KGE, so our results are unaffected by this, but for clarity of comparison between sites we have used a consistent colour scale on all KGE plots based on the standard thresholds.

---

[*] n.b. Both NSE and KGE are uncorrected for the magnitude of the variability of the observations $\sigma_{obs}$, (see Suppl. Info).

[**] n.b. KGE without the penalty terms (in $\mu$ and $\sigma$) reduces simply to Pearson's correlation coefficient $r = \frac{cov(Q_{sim}(t), Q_{obs}(t))}{\sigma_{sim}\sigma_{obs}} = \frac{1}{N-1}\frac{\sqrt{\sum_t ((Q_{sim}(t) - \overline{Q_{sim}})(Q_{obs}(t) - \overline{Q_{obs}}))}}{\sigma_{sim}\sigma_{obs}}$.




| | Fraction of gridcell inundated | |
|---|---|---|
| Site | GIEMS | JULES-CaMa-Flood |

Neotropics

Amazon

Pantanal



West Paleotropics

Niger Inland Delta

Sudd

Congo

Fraction of gridcell inundated (GIEMS)

Fraction of gridcell inundated (CaMa-Flood)





**Figure 1**: Fraction of gridcell inundated (in addition to water contained in channels and watercourses, which are not shown) in each study area. Superposed lakes and reservoirs are from the Global Lakes and Wetlands Database (GLWD) Lehner and Döll (2004). Resolution is 0.25° in both latitude and longitude (n.b. the Tonlé Sap is our smallest wetland, therefore the gridcells are relatively large in that plot). View window extent is taken from references in Table 1. Cities with populations >100 000 are shown **(SimpleMaps, 2019)** for view extents up to 2 000 000 km². Data shown are an average for 1992-2014 from *GIEMS-2* observations (left) and equivalent *JULES-CaMa-Flood* simulations (right).






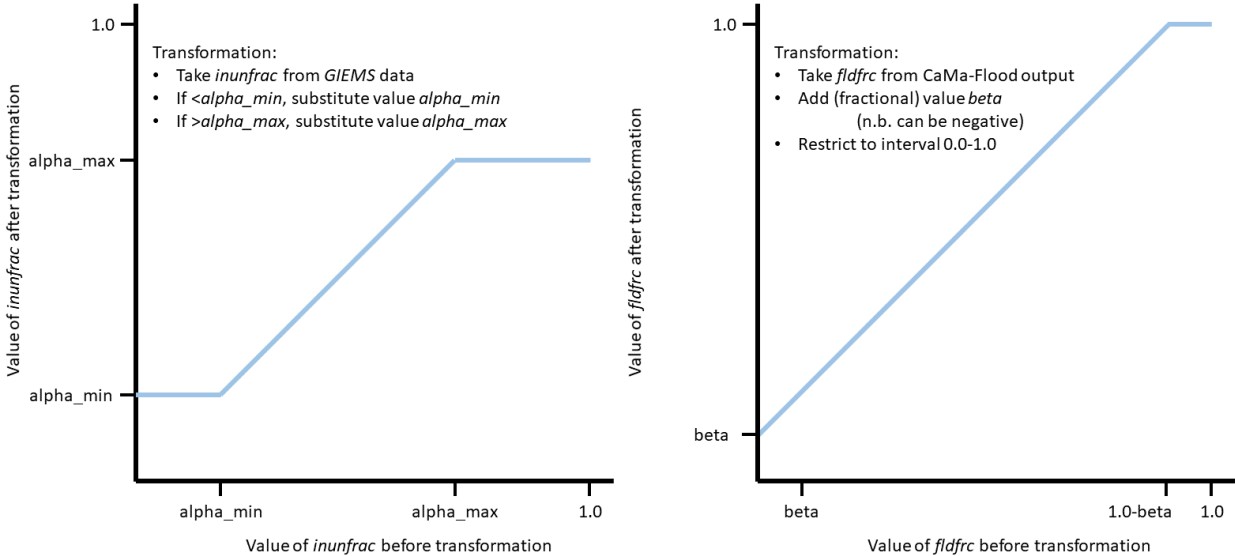

**Figure 2**: Transforming the *GIEMS* inundated fraction (*inunfrac*) data (left) and *CaMa-Flood* output flooded fraction (*fldfrc*) variable (right). Note that values *alpha_min = beta* =0.0 and *alpha_max* =1.0 are equivalent to making no modification.






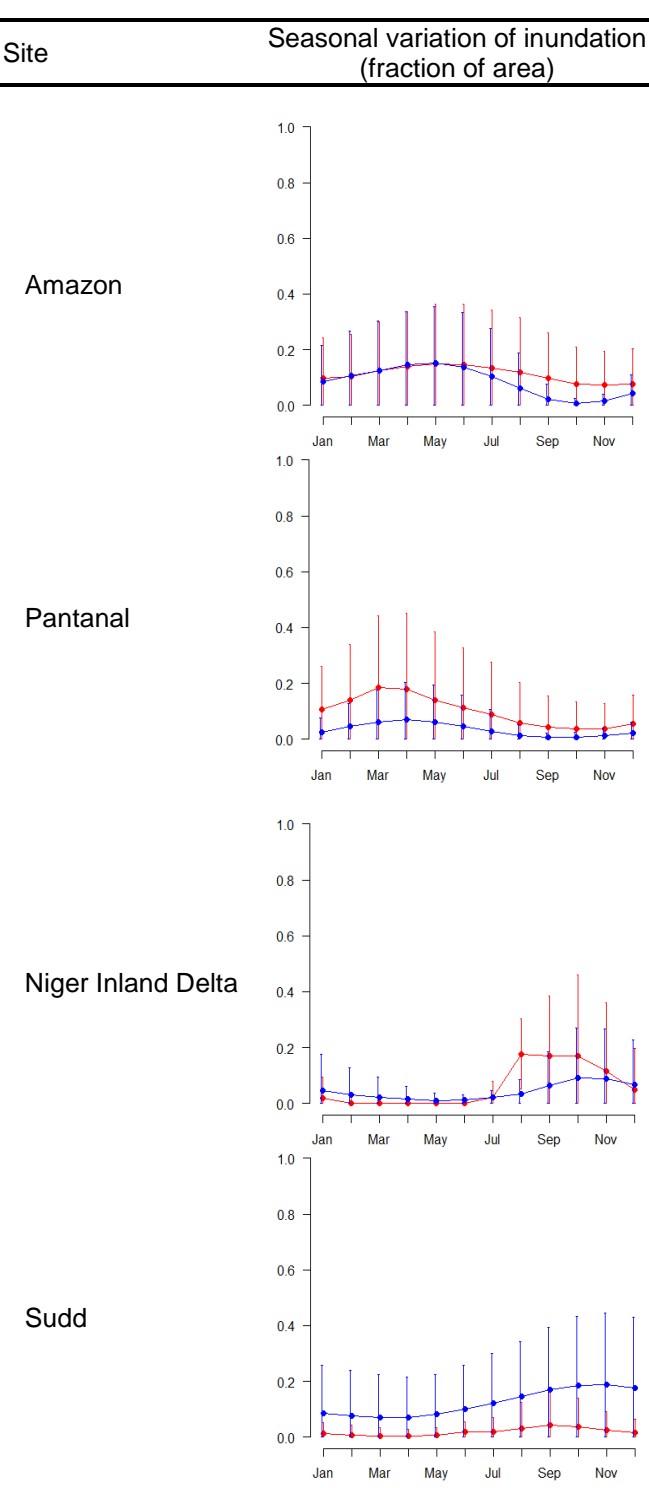



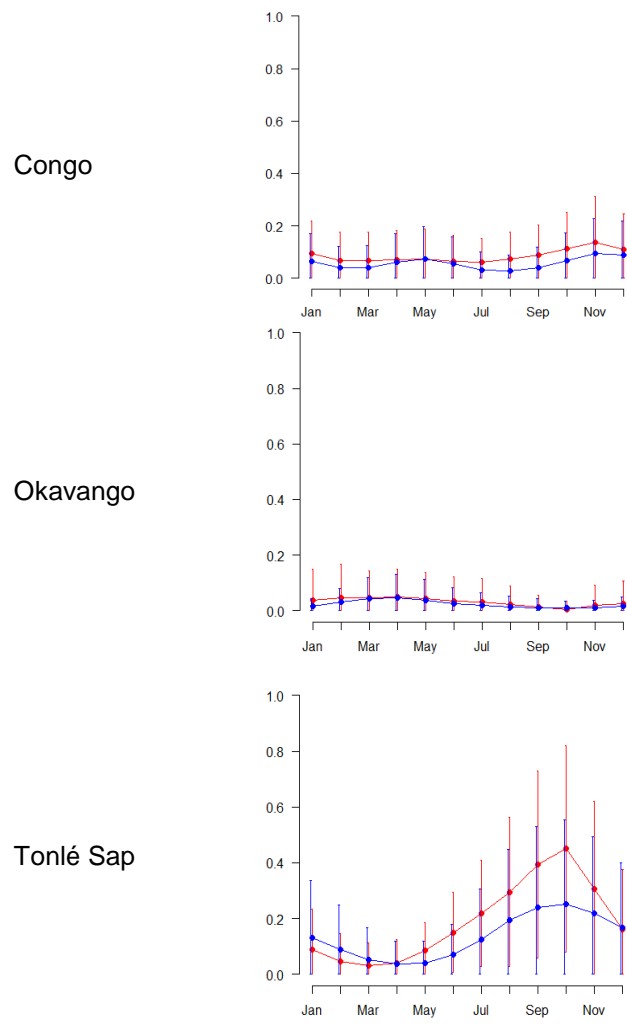

**Figure 3**: Seasonal variation in inundation across the study wetlands, averaged across the years 1992-2014: Red = Observations (*GIEMS*), Blue= Simulated (*JULES-CaMa-Flood*). The three main tropical zones are not shown because they include areas both north and south of the Equator.





| Site | Nash-Sutcliffe Efficiency, NSE | Kling-Gupta Efficiency, KGE |
| --- | --- | --- |
| Neotropics | | |
| Amazon | | |
| Pantanal | | |
| West Paleotropics | | |

Niger Inland Delta

Sudd

Congo

Okavango



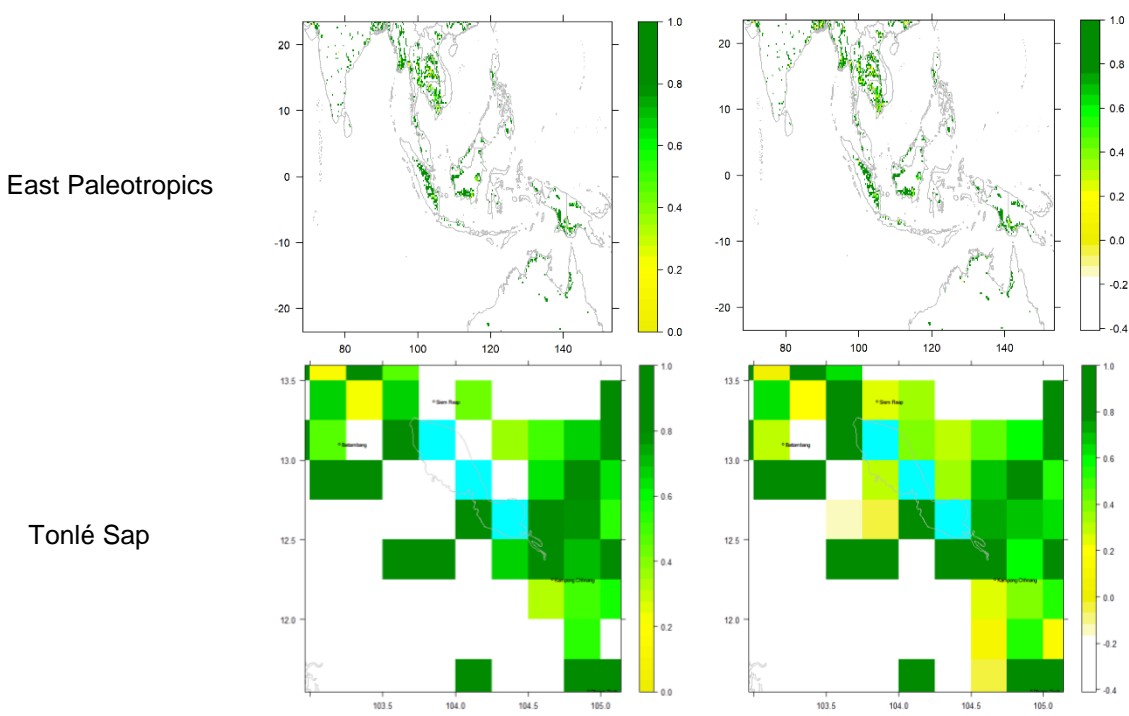

**Figure 4**: Mapped values for efficiency statistics based on inundated gridcell fraction, averaged across the years 1992-2014
(with *alpha_min*=0.0, *beta*=0.0 and *alpha_max*=1.0) (white indicates no value could be calculated).





| Site | Nash-Sutcliffe Efficiency, NSE | Kling-Gupta Efficiency, KGE |
|------|-------------------------------|-----------------------------|
| Neotropics | | |
| Amazon | | |
| Pantanal | | |



West Paleotropics

Niger Inland Delta

Sudd

Congo



**Figure 5a**: State space plots for evaluation statistics based on inundated gridcell fraction, calculated from varying parameters
*alpha_min* and *beta*, with panels showing values of *alpha_max*. Each point is the mean of all NSE or KGE values, averaged
both over time (years 1992-2014) and over the wetland region concerned (white indicates no value could be calculated).

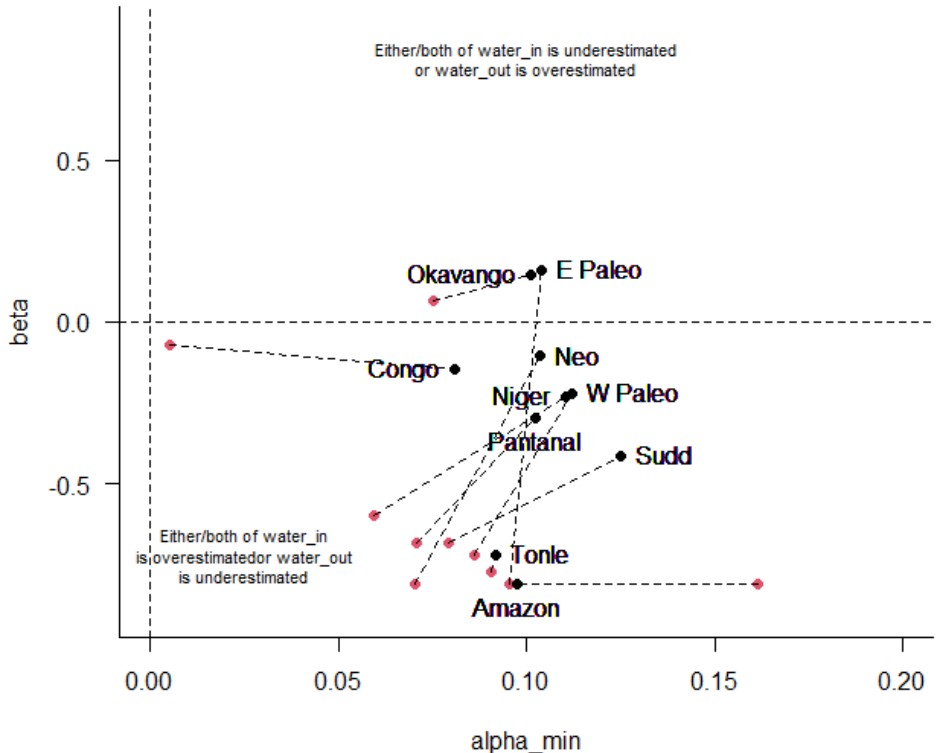

**Figure 5b**: Summary of plots in Fig. 5a. Optimal values of *beta* and *alpha_min* are shown (referred to as *beta_opt* and *alpha_min_opt* in the text), calculated as the centroids of the maximal region on the KGE plots (black) or NSE plots (red) for each site (with *alpha_max*=1.0) from Fig. 5a. On this plot, we define *water_in* = (channel + surface + subsurface inflow + precipitation) and *water_out* = (infiltration + evaporation). Note that clear maxima were not present for all case studies for NSE (Fig. 5a), but when present they are shown connected to the equivalent maxima for KGE.

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
