# Peer review of "Inundation prediction in tropical wetlands from *JULES-CaMa-Flood* global land surface simulations"

_Hydrology and Earth System Sciences, 2021_

## Author Comment (AC1)

**Online discussion**

**1** RC1: **'Comment on hess-2021-109'**, Anonymous Referee #1, 26 May 2021

Toby Marthews et al., conducted a global simulation of inundation areas with CaMa-Flood hydrodynamic model and was driven by JULES land surface model's runoff outputs at 0.25 by 0.25 degree resolution. They compared the simulated inundation areas against Global Inundation Extent from Multi- Satellites database version 2.0 (GIEMS2) dataset over several major inundated regions across the globe. They also tried to bias-correct the model simulated inundation area with simple transformations. Below are my specific comments.

- The major contribution of this analysis is better understanding of CaMa-Flood model biases, and the value of this work is so limited to the CaMa-Flood model/JULES model themselves. Little insights could be gained to better understand the mechanisms/processes underlying the regional hydrological cycle and water balance.

We thank the reviewer for their positive comments on our paper, however we dispute that our results are only relevant to the JULES/CaMa-Flood modelling community. We have added the following to our conclusions in section 4.4: "These comments are not only relevant to *GIEMS-2* and *JULES-CaMa-Flood* data: all satellite-based inundation data have biases that may be assumed to be very similar to those inherent in *GIEMS* data, and all model predictions of inundation have biases and uncertainties presumably similar to those that are in *JULES-CaMa-Flood* predictions (Dutra et al., 2015; Liang and Liu, 2020; Parker et al., in prep. 2020; Saunois et al., 2020), so we believe that our results and analysis provide a blueprint for users of other model/observational data on how they might assess and account for these types of bias in their own data."

- Furthermore, the understanding of CaMa-Flood model bias was also limited to how it is biased but little was known about why CaMa-Flood has such bias. Which specific process is responsible for the bias?

We very much agree that this is a very interesting question, but we have to suggest that we simply do not have the space in a journal article to discuss this in any depth. For example, it may be the case that a large proportion of CaMa-Flood's bias stems directly from its use of the local inertial approximation to the Saint Venant equations. Or perhaps that contributes very little and it is something to do with the general calculational approach taken by the model. The only way to interrogate this and decompose the bias according to model subroutine is to be able to do repeated simulations of CaMa-Flood with various physics options turned on and off. Although this was not planned or possible for the current study, we are currently working on a follow-up paper where we do precisely this. Only with this 'decompositional' data can we address these questions, and therefore in this current paper we have restricted ourselves to the slightly more general question of how CaMa-Flood behaves as a whole (i.e. not its individual components).

- In the methodology section, it is clear that JULES provided runoff outputs. However, it is not clear how accurate JULES runoff was. Although JULES runoff evaluation was published before, as the major driving variable of CaMa-Flood model, it's still worthwhile to e.g., add a full paragraph to summarize JULES' runoff at a global and regional scale (particularly the major inundated regions used in this study).
- Also, it will be great to have a full paragraph in the discussion section to discuss the contribution of runoff bias to the CaMa-Flood simulated inundation area bias.

Thank you for these comments, but this analysis of the JULES runoff output and a comparison between it and GRDC observations was carried out by Arduini *et al.* (2017) as part of the EU *eartH2Observe* project. Our study does rest partly on this verification, of course, but because this analysis has been published and the runoff data fields are available for public

download, we believe that it would not be appropriate to summarise this work here and it would detract from the focus of this paper, which is inundation rather than runoff.

50    It is always challenging to present results from large, sophisticated modelling frameworks: the outputs data fields have uncertainty that is contributed from multiple sources, not least all the individual stages in the modelling sequence (e.g. CaMa-Flood converts runoff into inundation extent, so of course runoff uncertainty is an element of the output uncertainty in inundation, but our runoff uncertainty may also be decomposed into uncertainty in JULES and uncertainty in the climate input data, and the climate data may be further decomposed). I discussed these different contributions at length in my paper last year
55    Marthews *et al.* (2020) where I analyses a wider set of eartH2Observe data.

However, in this particular paper we have taken our focus as inundation extent data and therefore we have been very cautious to restrict our discussion to issues that are relevant to this topic only, and from this point of view we do not believe it is relevant to reanalyse the eartH2Observe runoff data, or repeat details of the analysis ably carried out by Arduini *et al.* (2017).

60    • Again a more detailed explanation of the CaMa-Flood model ( inputs, outputs, major equations, hypotheses, advantages, disadvantages) is needed in the methodology section, although CaMa-Flood model description paper was published before.

Similarly to the last point, for reasons of space we have avoided adding in a full description of the CaMa-Flood model (or the
65    JULES model): these are, as the reviewer points out, published already and therefore we believe it is more correct for us to refer to these papers at the appropriate points.

• The results section needs a big refinement and explains more in detail (quantitatively). The current version (five short paragraphs) only scratches the surface of CaMa-Flood model results. Need more quantitative details about the analysis
70    of e.g., seasonality, interannual variability, spatial distribution, maximal inundation extent, functional relationships between inundation and environmental factors, and so on.

The reviewer is absolutely correct to request some refinement of these sections. We believe now that we had presented our results in a slightly confusing way, with some 'Results' text in the Discussion and some 'Discussion' text in the Results. We
75    have rewritten several paragraphs in this area now and we hope that this has greatly improved the readability of the paper.

• Discussion section, the bias in the inundation area needs to be mechanistically attributed to multiple relevant factors (e.g., precipitation, runoff) first before the bias-corrections so that one could learn why CaMa-Flood was biased and provide insights into how to bias correct the model through improving model structure, input data, parameterization
80    scheme and so on in the future.

We agree with this comment in principle, but with two provisos: Firstly, in addition to attributing uncertainties correctly to different parts of the modelling chain in order to identify points that may be improved, we must also bear in mind that the observational data itself also has biases and at the beginning we cannot know for sure which of these are the greater (it is
85    possible that CaMa-Flood is getting it broadly correct and the differences we see compared to GIEMS are because of issues such as vegetation cover or cloud cover). Secondly, we are aware that we are at the end of a modelling chain here: inundation extent depends on runoff (via CaMa-Flood in our case), which depends on precipitation and other factors (via JULES), which depends on previous climate state (via GCM/RCM) which itself depends on other model setup and assumptions. There is not enough space in a journal article to review all of these relevant factors, and we have been very careful to define the scope at
90    the start of this paper to include the last of these steps only. The topics we have selected and discussed in section 4 are, we believe, the most immediately relevant topics that will be of interest to a reader who may themselves be trying to predict inundation extent from supplied runoff data (a common situation nowadays).

• Discussion section, the bias correction (based on alpha min, alpha max, and beta) was empirical and may not be valid
95    if the bias was nonlinearly related to the space, time, and magnitude of the inundated area. In order to better justify the bias correction function, an analysis of the bias structure (across time and space) could be helpful.

The reviewer is completely correct to suggest that the bias may be nonlinearly related to aspects of the inundated area, however this does not invalidate the approach that we have taken: any nonlinear function may be approximated by a linear one and in the same way our bias-correction estimates form a first approximation of the real bias that does exist within the data under analysis. We have not analysed the bias structure because we do not have any data on the bias structure: we believe that it would very much be speculation for us to engage in this.

- Abstract, the second half of the abstract needs more quantitative results and deep implications. The last sentence is not convincing, since this study did not provide data, it was a model-data comparison study.

We have reported quantitatively our main results in the Abstract (apologies for having missed this!) and we have rephrased the last sentence of the Abstract to state that it is information on the biases in the data that are useful and timely.

Overall, we would like to thank RC1 very much for these results and for his/her time spent on our paper: we very much appreciate this feedback and we know that it takes significant time to undertake a thorough review like this.

Toby Marthews et al.

**Citation**: https://doi.org/10.5194/hess-2021-109-RC1

---

## Author Comment (AC2)

**Online discussion**

**1** RC2: **'Comment on hess-2021-109'**, Anonymous Referee #2, 27 May 2021

General comments:

This is a potentially interesting study comparing JULES-CaMa-Flood simulation output with a global dataset of inundation extent for different selected wetland regions.

Specific comments:

- There are 10 case study wetland regions, but the study spends insufficient time on most, if not all of them.

We thank the reviewer very much for their interest in the topic of our paper. We were very much hoping that submitting to *HESS* would elicit wider interest in our work on wetlands and we are very glad that this has turned out to be the case.

It is difficult to respond adequately to this request: all our case study wetlands are individually so well known that entire books have been written on many of them. Given the space limitations of a journal article - and our global perspective - we restricted our general site information for each case study area to a few brief lines in Table 1, where appropriate references have been given. We agree that we could have included a little more information on each wetland, but even devoting a paragraph of introduction about each wetland would have comfortably put us over our word limit for the paper, at the same time as opening us up to criticism that the information was superfluous because already published elsewhere. After much thought, we restricted ourselves to the minimum information on each wetland that was required for the reader to understand and interpret our results, which is the location and size of each (as given in Table 1), and in Table 1 we have collected together good references for each region through which the reader may access much more information about each very unique environment.

- The results section needs improving. The results within and between the regions studied should be compared in quantitative terms. The main/most important findings should be identified and highlighted.

Having considered the text in Results and Discussion, we believe that the problem identified (correctly) by the Reviewer here is that certain statements in our Discussion should have been placed in Results. We have revised the text that refers to Figs. 6 and 7 (formerly Fig. 5) and we hope very much that the new balance between Results and Discussion is much more readable than before.

- There are insufficient insights presented in the discussion. Can the errors and biases (based on the NGE, KGE, alphas and beta) at different locations be better attributed to the quality of JULES versus JULES-CaMa-Flood simulation? Are they predominantly a result of model structure, parameter or forcing errors? To what degree is uncertainty in the remote sensing data responsible? How do climate, season, and hydrotopograhy factor in? The authors must relate their results/findings with existing knowledge of earth system/hydrological processes of the different wetland regions studied.

The reviewer is disappointed that we have not provided a complete explanation of the biases we have detected in terms of physical variables (e.g. hydrotopography) and this is completely fair (and we share the disappointment!). However, in this paper we must restrict ourselves to conclusions that are justified by the data we have collected and the analysis that we have carried out. Ultimately, even though we have carried out a fairly sophisticated analysis, we are comparing only one variable (inundation extent) and this imposes hard limits on how much we can conclude. We could have included much more in the Discussion about what we believe to be the reasons for, e.g. the spatial displacement of inundation we detected in the Niger

Inland Delta or the apparent overestimation of water output from the Okavango swamps. These sections would perhaps have
made our Discussion more 'insightful', but we believe that we would have been speculating because we do not have additional
observational data to reinforce any such discussion. It may be of interest that we are currently working on a follow-up paper
where we attempt to identify mechanisms of inundation in a much more detailled way, but this requires more model runs (with
physical processes turned off/on) that we could not carry out for this study.

We have discussed several topics relevant to the hydrodynamics of wetlands in sections 4.2 and 4.3, linking back to previous
work in the Amazon. However, there are surprisingly few papers that have compared observed and predicted inundation extents
with robust protocols similar to our study, so we do not have many previous papers to compare to. Additionally, most global
studies are carried out at spatial scales of 1º or 2º or even coarser scale and report in terms of zonal means or meridional means,
which are measures that average over enormous areas of land and we find to be only peripherally useful at our spatial scale of
analysis. The studies we have referred to in the Discussion are the most relevant and comparable studies to ours, assessed after
a much wider literature search on our part. If this reviewer has particular studies that he/she would like to suggest that we have
perhaps missed in our search then we would very much like to hear about them, but given our current data and analysis we
believe this Discussion goes as far as our data permit us to go.

- The authors propose the alpha and beta parameters as indicators of model bias in the simulation of evapotranspiration
  and infiltration. However, these cannot be expected to be constant over time. Additionally, the NSE and KGE were
  calculated over the full temporal domain. Can the authors be confident that the high performing parameters remain
  valid at a different time? Additional analysis is warranted to investigate this.

We understand the drift of this comment to be a concern that by averaging over time (as required by the definitions of NSE or
KGE), we have lost the information on any variability over time. This is certainly true, but is a consequence of the definitions
of these statistics. The reviewer is absolutely correct that we cannot expect these indicators to be constant in time and we did
state this in our Methods ("our match statistics were dominated by the regular (seasonal) and irregular cycles occurring at
points where inundation was not constant").

During the analysis of this paper we attempted three different analyses relevant to this point, each with their own problems
and drawbacks, which may be explained by imagining a series of inundation fraction numberscovering a certain time period
P:
1. We looked at the spatial variability across pixels, but with each pixel's data averaged over the whole period P
2. We looked at the seasonal cycle for each wetland, which involved averaging across the spatial extent of the wetland
   and then compiling each month to acquire monthly averages (all the Aprils, all the Mays, etc.).
3. Looking at a seasonal cycle for each individual pixel

Note that studies based on river gauge data can only attempt approach #2 because they do not have spatial data (and even the
gauge data itself cannot be considered a mean of the wetland's dynamics because it is biased towards the dynamics immediately
around the gauge itself): Approach #1 especially is a high point of novelty of our study: the vast majority of wetland studies
are still based on gauge data only and we are not aware of any other study that has been able to take this spatially-explicit
approach.

Having attempted all of these, we quickly discarded #3 because of wetland fluctuations: if the inundation fraction goes to zero
(i.e. a particular point finds itself temporarily outside the wetland) or 100% (i.e. the central lake that many wetlands have
extends outwards) then our NSE and KGE statistics register –Inf, zero or one values. These values then outweigh all other
values in a straight averaging scheme. The usual solution is to mask out all points that are peripheral to the wetland area in this
way, but with many wetlands this gave us no remaining points at all. Partly, this is because of the recognised deficiencies of
these standard statistics (which we reviewed in Table 2), but effectively it meant that we had to discard this option.

Our results presented in Fig. 3 (formerly Fig. 1) and in Fig. 5 (formerly Fig. 4) are examples of approach #1, the results in Fig.
4 (formerly Fig. 3) are from approach #2 and there is much to be learnt from both analyses about the spatial and temporal
variability of these wetlands. Please note that because NSE and KGE require comparing time series, they can only be calculated
from approach #1.

We did consider stratifying the data into two periods of 11 years each and comparing 'before' (1992-2003) and 'after' (2003-2014) NSE and KGE values - and we did in fact carry out this analysis – but we found the conclusions we might have drawn from this were statistically unsafe. For example, what would it signify if we found a 10% increase in KGE for the Congo between the before and after periods? It could perhaps suggest climate change, but we could not exclude the possible effects of other social change in the local area (e.g. for the Congo, our 'before' period ended approximately at the end of the Second Congo War). Despite having found some interesting trends between 'before' and 'after', we decided not to include these results in the paper for essentially statistical reasons.

A similar argument applies to the bias-correction parameters *alpha* and *beta*. We agree that we do not expect these to be constant over time, however the nature of the statistics used required a time period for calculation and the period 1992-2014 was the longest period for which we had the required data. However, we do also agree that we have not mentioned this important point in the paper, therefore we have now added the following text to the end of the Results: "Finally, we note the specificity of our results to the time period 1992-2014. Carrying out this analysis for an earlier or a later period would certainly yield different estimates of NSE, KGE, *alpha_min*, *alpha_max* and *beta*. However, we suggest that without significant climate change, or perhaps significant anthropogenic modification of the wetland area concerned, the values of these statistics should remain similar to the values calculated here.".

Technical comments:

- Abstract: the final sentence of the abstract claims "This study provides timely data". However, it is unclear from the manuscript what part of the results/findings this "data" is referring to. The alpha and beta parameters are possibly only useful for JULES-CaMa-Flood-GIEMS users.

This sentence has been rephrased to state clearly that it is information on the biases in data that are useful and timely.
    Even though we have been careful to restrct our conclusions to GIEMS and JULES-CaMa-Flood data only, we do not believe that these are the only communities who would find use in these data: all satellite-based inundation data has biases that may be assumed to be very similar to those inherent in GIEMS data, and all model predictions of inundation have biases and uncertainties presumably similar to those that are in JULES-CaMa-Flood predictions, so we believe that this paper also provides a blueprint for users of other model/observational data on how they may assess and account for some kinds of bias in their own data. We have added this comment to the conclusions in section 4.4 as well now.

- Abstract: in line with the earlier recommendation to investigate all the different regions more thoroughly "(including the Sudd, Pantanal, Congo and Amazon)" should be removed.

Parentheses removed as requested

- Line 35: what is being referenced to in the cited reference Saunois et al 2020 is unclear.

Reference removed

- Line 100: "Most hydrological models are run uncoupled from the atmosphere and are therefore reliant on the availability of good precipitation and other atmospheric driving data." – the first part of this statement is inconsequential. Even if hydrological models were run coupled with atmospheric models, a high level of error from the simulated precipitation is still expected.

Perhaps this might be considered a point of terminology only, but we do see a significant difference here. In an uncoupled framework, the user acquires precipitation data from (usually) a published source (e.g. MSWEP data) and the uncertainty in the precipitation is considered a form of data uncertainty. Conversely, in a coupled model context the precipitation is calculated by one of the component models of the coupled framework (the GCM/RCM) and any uncertainty in this is therefore considered

to be internal, i.e. model uncertainty. The point being made in this paragraph (and in my 2020 paper referred to here) was about data uncertainty *vs.* model uncertainty. We would completely agree that precipitation has notoriously high uncertainty whatever the modelling framework, but sometimes this is external and sometimes it is internal and this is what was referred to here. We have changed "availability of good precipitation and other atmospheric driving data" here to "the availability of independently-sourced precipitation and other atmospheric driving data" to "availability of high-quality precipitation and other atmospheric driving data obtained from independent sources".

- Include a study area figure at the global scale to adequately introduce the wetland regions and discuss their differences in major processes/controls. This will remove the need to refer to a few of these regions as "the three tropical zones", which can be confusing for the reader.

Thank you very much for this suggestion: we have now included a new Fig. 1 that shows the locations of all 10 study areas, from which we feel it is now much more clear that 3 are larger than the others and are more correctly referred to as "tropical zones" rather than "wetlands".

- Lines 201, 215: references to figures from the results section within the methods section should be removed.

The reference to Fig. 1 on line 201 has been removed, but the reference to Fig. 2 on line 215 has been retained because this figure was indeed truly a 'Methods' figure. In order to clarify the text, the figures have been renumbered so that the 'Methods' figure comes before the 'Results' figure.

- Line 222: "We therefore calculate spatial matching statistics across all case study areas" - it is unclear what is being meant here by spatial matching statistics.

Thank you for pointing this out: we indeed had only used this term once and we agree it is not especially informative. The setnence has been rephrased with "appropriate statistics" instead

- Lines 228-229: The evaluation metrics nRMSE, r, RMSE were not introduced in the methods, nor were their results presented.

We did state in our Methods section that the results of our RMSE and Pearson's *r* calculations were not presented. From feedback during the writing of this paper, there is a wide expectation that these statistics should be applied in an analysis such as ours, so we should mention that we have done so, but we found that there were no additional conclusions to be drawn from the RMSE and *r* plots that could not be drawn from the corresponding NSE and KGE plots (and the trends were always more clear on those plots too). Statistically, this is simply because NSE and KGE include (i.e. may be decomposed into) the definitions for RMSE and Pearson's *r* (as we describe in Table 2). In summary, we need to mention RMSE and *r* briefly because of the widespread expectation that they should be applied, but because NSE and KGE include these statistics, there were no additional results to be learned from these plots and we did not wish to mention them more than briefly (especially because presenting these results as well would have meant including an additional Fig. 5 (formerly Fig. 4) and Fig. 6 (formerly Fig. 5a) based on these statistics).

- Line 238: "However, these statistics are not capable of measuring some aspects of the flow regime that are important from the point of view of allowing us to divide out the different sources of inundation in our study wetlands" is unclear.

We have rephrased this sentence with an extra few lines explaining the significance of spatial displacement of inundation.

[Figure]

PDF online version in HESSD                                     Version submitted (Word Doc)

- Figure 1, 4: the results for regions with a larger spatial domain are difficult to see.
- Figure 5 is blurry

200

We apologise for the difficulty here: it is difficult to present continental-scale results in an easily-accessible figure in this way because we are limited in terms of available space. However, we do note that we have included high-resolution images in these figures and the user can zoom into the figure to see additional detail. This will also be possible in the online version of these figures as well.

205    We also note that in the PDF versions of these figures available for review, the quality of these images seems to have been downgraded (see excerpt from Fig. 4 above). This will not occur if the paper moves to HESS for full publication and the images will be much sharper.

- Line 237: "within the borders of the wetland itself" – at/near the wetland boundaries?

210

Apologies for not having been clear here: we meant "across the wetland" and have rephrased this sentence appropriately.

- The authors' conclusion in lines 284-288 is poorly supported.

215   The conclusion in question here was our statement that "The visible maxima on these state space plots provide a best estimate of the optimal values of these parameters, with these optima differing markedly between our wetland study areas (Table 1). A notably higher value for NSE or KGE for a particular combination of *alpha_min*, *beta* and *alpha_max* would identify a consistent bias in either the model predictions or the observations (or both)."

   With apologies, we believe that this statement is emphatically supported by our data: our search of the parameter space

220 values to obtain these visible maxima was exhaustive and therefore the conclusion that these are real optima is robust. Some of these optima were like the high point on a gently sloping hill rather than a clear mountain summit, but very clear optima on state-space diagrams are in any case rare in analyses such as this. The optima very obviously differ between our case study wetlands (as is visible from Fig. 6, formerly Fig. 5a).

   We would request reviewer RC2 to please be more specific as to what aspect of this he/she found unconvincing and we will

225 try our best to redress?

- Line 302: "If overbank flooding is underestimated in our simulation then the water within the river course (the Niger or White Nile, respectively, in these cases) will remain in the river and be taken downstream further than expected, producing a downstream wetland 'extension' that exists in the simulation results but not the observed (as we see in our JULES-CaMa-Flood outputs). – this needs to be better linked better to the study results, with examples.

This is an example of the process of spatial displacement of inundation. In response to another reviewer comment above, this has been explained more fully now in the text at lines 245-250.

- The manuscript is informal in tone and unfocussed at some parts, with longwinded sentences that make it hard to read. Additionally, there are:
  - acronyms undefined at first use e.g. CaMA, GIEMS, GLWD, WRR1, WRR2

We apologise for this: all these acronyms have now been defined at first use in the text.

  - use of biased/subjective words, e.g. "surprisingly", "sophisticated"

We have removed the one occurrence of "sophisticated", but we would like to retain the one occurrence of "surprisingly" in the first sentence of the Abstract. We are stating here that "reliable data on the extent of global inundated areas ... remain surprisingly uncertain" and we do believe that this is objectively surprising.
    Without exception, when we have discussed this paper with non-hydrologists in order to receive some feedback on the content of the paper, the first reaction has been surprise that information on inundation extents should not be extremely well-known. It is unclear what this assumption is based on, but we suspect simply because it seems such an easy thing to measure (a depth gauge) and has been measured in some locations for so long (e.g. the Ancient Egyptians measured the inundated extent of the Nile river), so people assume that in the 'modern' world we should have universal data on this quantity. Once the difficulties are explained (measurement in ungauged catchments, real time requirements, satellite uncertainties) then it generally becomes clear, but even for hydrologists (and, we assume, the audience of *HESS*) the current state of global data is almost universally a surprise. To remove this word at the start would imply that our results are of far smaller importance than we believe them to be, so we would like please to keep it.

  - overstatements, e.g. "widely used" (relative to the few references cited)

We have been trying to avoid over-referencing in this paper. When we stated that CaMa-Flood was a widely-used model, we referred to Hoch & Trigg (2019) and Zhao et al. (2017), both of which are good examples of CaMa-Flood being used but also give references to other studies where the same model has been used for more than a decade now. We believe that either one of these references on their own would be sufficient to support this statement, and we could have added in more examples, but we did not feel that more references were warranted in this case.

  - excessive use of brackets and italicized phrases that highly disrupt the flow

We are not aware of any excessive use of parentheses or italicised terms: the topic under discussion is relatively technical, which unfortunately requires use of technical jargon (on the use of which this reviewer has corrected us a few times, for which we are very grateful!) and we have tried as far as possible not to make unsupported statements that may be interpreted as subjective, which requires a fair number of citations, but we do not feel that these have been excessive. We have been through the text completely and have attempted to improve the flow of the text at certain points, and we hope very much that this makes the paper read a little easier.

  - generic/blanket statements such as: "We found that our simulated inundation extents (from the CaMa-Flood model, driven by JULES runoff data at 0.25° resolution) sometimes compared very closely to our observed data (from GIEMS satellite-based data), but at many points there were divergences", and "The spatial

displacement of inundation prediction downstream from observed inundation visible especially in our results for the Inner Niger Delta and the Sudd (Fig. 1) is a result of over- or under-estimation of overbank flooding upstream." There are multiple occurrences throughout the manuscript including in the abstract.
The overall readability must be improved.

280

We have tried to improve the readability of the text throughout. Addressing the two examples highlighted here, these are direct statements of fact, however: our simulated extents did indeed match very precisely at certan points, but not at others (here we were introducing the results and making the general comment that there was variability in how well the modelled extents matched observed) and the second is a statement about spatial displacement of inundation that is indeed usually a result of
285    upstream effects. We fully admit that if our intended audience were hydrologists only then these would be almost superfluous statements, but the readership of *HESS* is wider than just hydrologists so we believe that statements like these are necessary in order to introduce concepts that may not be familiar.

Overall, we would like to thank RC2 as well very much for these results and for his/her time spent on our paper: we understand
290    that it is quite an imposition to do a review and are very grateful for the feedback received.

Toby Marthews et al.

**Citation**: https://doi.org/10.5194/hess-2021-109-RC2
295

---

## Author Response (AR2)

UK Centre for Ecology & Hydrology (UKCEH)

Maclean Building,

Wallingford,

U.K.

16th February 2022

Dear Prof. Gentine,

**Re:** Marthews *et al.*

**Inundation prediction in tropical wetlands from JULES-CaMa-Flood global land surface simulations**

HESS submitted paper https://editor.copernicus.org/HESS/ms_records/hess-2021-109

Very many apologies for the delay in responding to the two reviews received for the paper above. We were only able to see the final reviewer response on 1st February.

Firstly, we would like to thank you very much for sourcing the two Reviewers for this paper: their comments have been extremely insightful and identified several shortcomings to the paper. We are very grateful to have had the opportunity to address and correct these during the discussion period.

We have revisited all parts of the manuscript in the light of these comments and we truly believe that the paper is now very much stronger than our original submission. We have had two rounds of reviewer comments to date, and extensive improvements have been made to the paper both last July and this month. These include the insertion of new analyses and figures (e.g. Fig. 1 which was not in our original submission) and the complete revision of three sections in the Methods and Conclusions that we hope clarify the overall presentation of the paper and argue important points in full that were only briefly mentioned in our original submission.

We hope that the revised version of the paper attached and the comments below answer all remaining reviewer concerns.

Very many thanks again for the time that you have spent considering our paper for publication.

Best regards,

Toby Marthews *et al.*

*Response to specific reviewer comments received*:

Reviewer #2, although he/she submitted the most comments during the first round of review last July, is now very happy with the paper (labelling it "Excellent" for scientific quality in the MS records). He/she has requested a few extra changes in their latest report, which have all been implemented without any modification (reordering figures and rephrasing a few sentences).

For reference, a point-by-point response is included at the end of this letter below (Report #2).

Reviewer #1 has raised several important points that were not fully covered by our responses back in July 2021. He/she is essentially concerned that we have not considered the various model biases inherent in our analysis in sufficient detail, both at the JULES stage and the CaMa-Flood stage of our modelling workflow. As a result of this, my coauthors and I have been through the draft paper in detail again. We did indeed consider model bias and uncertainty throughout our analysis, but on reflection we do agree that this was not made sufficiently clear in our original submission and we have therefore made substantial changes to the new manuscript to address the reviewer's concerns. Essentially, Reviewer #1 made two requests:

- Reviewer #1 requested "In the methodology section, it is clear that JULES provided runoff outputs. However, it is not clear how accurate JULES runoff was. Although JULES runoff evaluation was published before, as the major driving variable of CaMa-Flood model, it's still worthwhile to e.g., add a full paragraph to summarize JULES' runoff at a global and regional scale (particularly the major inundated regions used in this study)."
  - In addition to the substantial improvements made last July, Section 2.2 in *Methods* has now been expanded to include two entirely new sections: 2.2.1 ("**Validation of land surface runoff**") and 2.2.2 ("**Validation of land surface inundation**") and in these sections we have given extensive details exactly along the lines requested by Reviewer #1. These new sections give summaries of the validation of both runoff and inundation data procured from our models.

- Reviewer #1 also requested: "Also, it will be great to have a full paragraph in the discussion section to discuss the contribution of runoff bias to the CaMa-Flood simulated inundation area bias." and again in a later comment "Discussion section, the bias in the inundation area needs to be mechanistically attributed to multiple relevant factors (e.g., precipitation, runoff) first before the bias-corrections so that one could learn why CaMa-Flood was biased and provide insights into how to bias correct the model through improving model structure, input data, parameterization scheme and so on in the future."
  - We have considered these requests and do agree that we needed to provide more detail on these important issues. We have expanded the Discussion to include a new section "4.1.2 **Quantifying bias in JULES-CaMa-Flood inundation predictions**" where we present the major

factors that we believe contribute to the bias we found in our analyses both in the model and observations of inundation extent. Our belief is that the key factor is whether a wetland may be considered to be groundwater-maintained or derives the majority of its flow from fluvial inundation. The values we have obtained for the statistics *beta-opt* for each wetland are the most indicative here and show clearly, we believe, in which wetlands we might expect the bias to be positive and in which we should expect the opposite (and we have illustrated this in Fig. 7). We have now included a thorough argument of the significance of this and its implications for the assessment (and future development) of models like *JULES* and *CaMa-Flood*.

We hope very much that these new sections form a sufficient response to the remaining concerns of Reviewer #1. As we have argued in the paper, we believe that this research makes a valuable and timely contribution to the current state-of-the-art in the model prediction of global wetlands: we are not aware of any comparable validation of observations *vs* predictions in this field and we believe that our results will, if published, effectively set the standard for future similar studies to follow.

Best regards,

Toby Marthews

UKCEH

*Report #2:*

Minor comments

- In abstract: timely information on inherent biases in inundation data -> inherent biases in inundation prediction and observation

Done

- The new figure 1 makes it clear there are different categories of the wetlands/wetland zones based on their scale. I suggest reorganizing the order of wetlands/zones in table 1 and figures 3-5 according to these categories. Additionally, please add latlong grid/ticks and labels, scale, legend. Revise caption - why "Example"?

Fig. 1 and legend updated (many apologies: I should have put in the scale and tick marks before). I have also reordered the rows in all the tables so that the wetlands are in progression of scale, as requested here (from regional to subcontinental to continental; from west to east).

- By convention, acronyms should be in brackets and full name spelled outside, rather than the opposite. Also, acronyms are still undefined in the abstract.

The text has been searched and the use of acronyms regularized.

- Add "and references therein" to Hoch & Trigg (2019) and Zhao et al. (2017).

Done

- Suggest to rephrase "perhaps the inundation is actually real but for some reason unobserved" to "perhaps inundation actually occurred but was unobserved"

Done - thank you for this!

- Please double check that all italicized terms representing model variables have been defined at first use.

Text has been searched to ensure this.

- Line 65, remove "="

Done